# Vesicular Stomatitis Virus Elicits Early Transcriptome Response in *Culicoides sonorensis* Cells

**DOI:** 10.3390/v15102108

**Published:** 2023-10-18

**Authors:** Stacey L. P. Scroggs, Edward J. Bird, David C. Molik, Dana Nayduch

**Affiliations:** 1Arthropod-Borne Animal Disease Research Unit, Agricultural Research Service, United States Department of Agriculture, Manhattan, KS 66502, USA; david.molik@usda.gov (D.C.M.); dana.nayduch@usda.gov (D.N.); 2Department of Entomology, Kansas State University, Manhattan, KS 66502, USA; edwardbird@ksu.edu

**Keywords:** vesicular stomatitis virus, VSV, VSNJV, negative-sense RNA virus, arbovirus, *Culicoides*, biting midge, transcriptome, immune response, zoonotic

## Abstract

Viruses that are transmitted by arthropods, or arboviruses, have evolved to successfully navigate both the invertebrate and vertebrate hosts, including their immune systems. Biting midges transmit several arboviruses including vesicular stomatitis virus (VSV). To study the interaction between VSV and midges, we characterized the transcriptomic responses of VSV-infected and mock-infected *Culicoides sonorensis* cells at 1, 8, 24, and 96 h post inoculation (HPI). The transcriptomic response of VSV-infected cells at 1 HPI was significant, but by 8 HPI there were no detectable differences between the transcriptome profiles of VSV-infected and mock-infected cells. Several genes involved in immunity were upregulated (*ATG2B* and *TRAF4*) or downregulated (*SMAD6* and *TOLL7*) in VSV-treated cells at 1 HPI. These results indicate that VSV infection in midge cells produces an early immune response that quickly wanes, giving insight into in vivo *C. sonorensis* VSV tolerance that may underlie their permissiveness as vectors for this virus.

## 1. Introduction

Vesicular stomatitis (VS) is a zoonotic disease caused by vesicular stomatitis virus (VSV, *Rhabdoviridae*, vesiculovirus), a single-stranded, negative-sense RNA virus that is transmitted by hematophagous insects like biting midges, black flies, and sand flies [1,2]. There are two serotypes of VSV, New Jersey (VSNJV) and Indiana (VSIV) [3,4]. VS primarily affects horses, but other livestock such as cattle, swine, sheep, goats, llamas, and alpacas can also become infected [3,4]. Clinical symptoms of VS include vesicular lesions on the mouth, naso-oral mucosa, teats, or coronary bands, loss of appetite, weight loss, and lameness [3,4,5]. These symptoms can be mistaken for those of foot-and-mouth disease (FMD). Once detected, VSV-positive premises can be placed under quarantine, which can have a major economic impact [6,7,8].

In the United States, *Culicoides sonorensis* biting midges (Diptera, Ceratopogonidae) are a known vector of VSV, showing non-lytic infection in all tissue types following ingestion of an infectious blood meal [9], and are able to transmit the virus efficiently to animals [10,11] and to other midges [12]. Female midges require bloodmeals for oviposition and are capable of three to four feeding/gonotrophic cycles during their lifetime [13]. Midges and other hematophagous vectors possess an innate immune system that responds to some arboviral infections. The primary immune signaling pathways in insects are the Janus kinase signal transducer and activator of transcription (JAK/STAT), Toll, immune deficiency (IMD), and RNA interference (RNAi) pathways [14,15,16,17]. Unlike other disease-transmitting arthropods, such as mosquitoes [18] and ticks [19], the molecular relationships between the *C. sonorensis* immune system and the pathogens it vectors are largely unknown. However, one prior study of epizootic hemorrhagic disease virus (EHDV) infection in midges demonstrated that antimicrobial peptide genes were upregulated at 36 h post ingestion of the virus in a blood meal, although it was not determined whether these effectors were aimed at the virus [20].

This study characterized the transcriptomes of VSV-infected *C. sonorensis* W8 cells in culture at multiple time points and hypothesized that VSV infection would suppress the midge innate immune response. Transcriptomes of VSNJV-infected and mock-infected *C. sonorensis* cells were deeply sequenced and differentially expressed unigenes (DEGs) were identified early in infection only, at 1 h post inoculation (HPI), while, interestingly, no DEGs were identified at or after 8 HPI. Several immune-related unigenes were downregulated (*SMAD6* and *TOLL7*) or upregulated (*ATG2B* and *TRAF4*) with VSNJV infection. These data indicate that VSNJV infection in midge cells elicits an immune response early in infection that quickly wanes even with continued robust viral replication.

## 2. Materials and Methods

### 2.1. Cells and Virus

*Culicoides sonorensis* W8 cells, generated from 1-day-old embryonated eggs (USDA, ARS, Arthropod-Borne Disease Research Unit, Manhattan, KS, USA) [21], were grown at 27 °C in Schneider’s insect media (MilliporeSigma, St. Louis, MO, USA) supplemented with 0.4 g/L of sodium bicarbonate, 0.0585 g/L of L-glutamine, 0.006 g/L of reduced glutathione, 0.03 g/L of L-asparagine, 18 μL of 10 mg/L bovine insulin, and 5% fetal bovine serum (FBS) [21,22], hereafter referred to as complete media. Vero MARU cells (Middle America Research Unit, Panama) were grown at 37 °C with 5% CO_2_ in 199E media supplemented with 2% FBS, 100U of penicillin/streptomycin sulfate, and 0.25 μg/mL of amphotericin B. The New Jersey serotype of VSV (bovine field isolate from 1982, USDA-APHIS, Ames, IA, USA) was grown at 37 °C with 5% CO_2_ in porcine epithelial cells (AG08113, Coriell Institute, Camden, NJ, USA) with Eagles MEM with Earle’s salts media (Sigma, St. Louis, MO, USA) supplemented with 2% FBS and 100U of penicillin/streptomycin sulfate.

### 2.2. Time Course Infection of W8 Cells with VSNJV

W8 cells were seeded onto 6-well plates at 2 × 10^7^ cells per well, two plates per time point (1, 8, 24, 48, 72, 96, and 120 HPI). The broad time points were selected to capture the full breadth of viral replication kinetics. For each time point, one plate was infected with VSNJV diluted in complete media for a multiplicity of infection (MOI) of 5 while the other plate was mock-infected with complete media alone. A total of six replicates were included per time point. Once the inoculum was added, the cells were incubated at 27 °C for 1 h with a gently rocking motion every 15 min. After which the inoculum was removed and the cells were washed twice with 2 mL of complete media before 2 mL of complete media were added to the wells. At each time point, the supernatant was collected, clarified by centrifugation (1500× *g*, 10 min, 4 °C), and stored at −80 °C. For the initial time point, the second wash was collected at 1 HPI and frozen at −80 °C.

### 2.3. VSNJV Quantification

Infectious VSV was quantified via plaque assay. Vero cells were seeded in 12-well plates at 2 × 10^6^ cells per well one day prior to infection. Clarified supernatants were serially diluted 1:10 in 199E media before inoculation onto the Vero cells. After incubation at 37 °C for 1 h with occasional rocking motion, the inoculum was aspirated and replaced with an overlay of 0.6% methylcellulose in 199E media. The plates were incubated for 3 days at 37 °C then fixed with a formaldehyde and crystal violet stain. After 1 h of incubation at room temperature, the stain was removed, wells were washed with water, and the plaques were counted. Viral titer was calculated as the number of plaque-forming units (PFU) per mL.

### 2.4. W8 RNA Extraction, Sequencing, and QC

After the supernatants were removed, the cells were lysed with Xml Trizol and RNA was collected per the manufacturer’s instructions (Direct-zol RNA Miniprep Plus Kit; R2070). RNA was stored at −20 °C. RNA quantity and quality were assessed via QubitFlex™ (Thermo Scientific, Waltham, MA, USA) and NanoDrop 8000™ (Thermo Scientific). Messenger RNA from the extracted RNA of samples for 1, 8, 24, and 96 HPI was purified using poly-T oligo attached magnetic beads. Purified mRNA was fragmented, and first-strand cDNA was synthesized via a random hexamer and completed using second-strand synthesis. A one-step end-repair and dA-tailing method was used to create 5′-phosphorylated and 3′-dAtailed cDNA fragments enabling direct ligation of Illumina sequencing adapters. cDNA was then size-selected and amplified by polymerase chain reaction (PCR). Multiplex libraries were sequenced on the Illumina Hiseq2500 platform with at least 40 million paired-end 150BP reads per sample. Low-quality sequences were removed by trimming with fastp v0.23.2 [23] (Q = 25) and sequence quality was validated with FastQC v0.11.9 (https://www.bioinformatics.babraham.ac.uk/projects/fastqc/ (accessed on 2 May 2023).

### 2.5. De Novo Assembly, Reduction, and Annotation from W8 Cells

Trimmed reads from each sample were concatenated and normalized to 50× k-mer (k = 25) coverage using Trinity’s in silico read normalization. Normalized reads were then assembled using the default Trinity pipeline v2.14.0 [24]. The resulting transcriptome was reduced using the default EvidentialGene tr2aacds v2018.06.18 [25] pipeline to filter out biologically irrelevant or redundant transcripts. Kraken2 v2.1.3 [26] was used with the NCBI nucleotide database (2 May 2023) to filter out human transcripts and separate out candidate VSV transcripts using Kraken Tools v1.2 [27]. This final transcriptome assembly was annotated using the default Trinotate pipeline v3.2.2 [28], combining evidence from Pfam-a HMM [29], Swiss-Prot [30], and UniProt [31] (Accessed June 2023). Gene Ontology (GO) terms and Kyoto Encyclopedia of Genes and Genomes (KEGG) terms were extracted from the best Swiss-Prot and Pfam hits.

### 2.6. Differential Expression from W8 Cells

Trimmed reads were pseudo-aligned to the final reference transcriptome coding domains of the VSNJV genome (NCBI accession MK613994) using Salmon v 1.10.1 [32] and then unigene counts were generated for each sample. Unigene-level abundance, counts, and length were imported to R 4.1.3 [33] using the tximport [34] package. Samples were grouped into a single factor based on the combination of treatment and time, represented in metadata. A negative binomial model (~Trt_Time) was fit and dispersion was visualized using Deseq2 [35] (Appendix A). Wald hypothesis testing contrasted infected vs. uninfected at each time point, with a test of adjusted *p*-value > 0.05 [36] and Log2(Fold-Change) > 0.58.

### 2.7. Statistical Analysis

Differences in log-transformed viral titers over time were detected using repeated measures ANOVAs with Tukey’s multiple comparisons tests. Differences in viral unigenes per million by gene were detected with two-way ANOVAs with Fisher’s multiple comparisons tests. Correlations between viral gene order and unigenes per million were detected with Pearson correlation coefficients. Statistics were conducted using GraphPad Prism v 10.0.2 (GraphPad Software, Boston, MA, USA). The principal component analysis, heatmap, model dispersion plot, and volcano plots were all produced in R [33] with packages GGplot, pheatmap, DESeq2, DEGreport, and Enhanced Volcano.

## 3. Results

### 3.1. VSNJV Rapidly Produces Infectious Virus W8 Cells

To evaluate the transcriptional response of W8 *C. sonorensis* cells to VSNJV infection at multiple time points, we infected or mock-infected six replicates each of W8 cells and collected cell supernatant and cell lysates at 1, 8, 24, 48, 72, 96, and 120 HPI. As seen in Figure 1, the infectious virus rapidly increased and peaked between 8 and 24 HPI (5.32 Log_10_ PFU/mL (0.22 SE) and 5.68 Log_10_ PFU/mL (0.11 SE), respectively) and then remained high until the conclusion of the experiment at 120 HPI (5.67 Log_10_ PFU/mL, 0.09 SE). As expected, the average viral titer at 1 HPI was significantly lower than that of all other time points (Tukey’s multiple comparisons adjusted *p*-value < 0.0001).

### 3.2. W8 Transcriptome Response to VSNJV Infection

Transcriptome analyses were conducted on RNA extracted from W8 cells infected with VSNJV or mock-infected at 1, 8, 24, and 96 HPI (six replicates per condition per time point). Gene ontology (GO) annotation analyses for all unigenes and time points identified the most numerous unigenes as components of the cytoplasm (18%), nucleus (16%), cytosol (13%), plasma membrane (11%), and membrane (10%) (Appendix A). The primary molecular functions were ATP binding (17%), metal ion binding (15%), and protein binding (10%) (Appendix A), while the primary biological processes were proteolysis (12%) and regulation of transcription by RNA polymerase II (9%) (Appendix A).

Unigene expression profiles from 8, 24, and 96 HPI showed no differences in clustering between VSNJV-infected and mock-infected cells (Figure 2). However, differences in unigene expression at 1 HPI between VSNJV-infected and mock-infected cells were detected by principal component analyses (PCA) (Figure 2 and Appendix A). PCA1 and PCA2 (Figure 2A) demonstrate that the samples cluster primarily by time post inoculation meaning the majority of the variance is explained by time. But two distinct clusters form with PCA1 and PCA3 (Figure 2B) by infection status at 1 HPI indicating that the unigene profiles differ by infection status only at 1 HPI. The pattern was also found using a correlation matrix (Figure 3) that demonstrates that the mock-infected and VSNJV-infected profiles from 8, 24, and 96 HPI were highly correlated by time point. The correlation of mock-infected and VSNJV-infected profiles at 1 HPI was present, but weaker compared with the later time points.

Using a cutoff of Log2FC = 0.58, or about a 1.5-fold change, 102 differentially expressed unigenes (DEGs) were identified at 1 HPI with 54 upregulated and 48 downregulated unigenes (Appendix A). No DEGs were identified at 8, 24, or 96 HPI. Half (51.0%) of the downregulated unigenes were suppressed between −1.0 and −2.0 log2FC in the VSNJV-infected cells. Of the remaining unigenes, 40.8% were downregulated between −0.58 and −1.0 log2FC and only 8.2% had log2FC values less than −2. The majority of the downregulated unigenes (73.5%) were able to be assigned putative annotations via homology to entries in the Swiss-Prot database. Similar to the downregulated unigenes, half (49.1%) of the upregulated unigenes were upregulated between −1.0 and −2.0 log2FC in the VSNJV-infected cells. The remaining upregulated unigenes were 2.0 log2FC or more upregulated in 35.8% of the unigenes and between 0.58 and 1.0 log2FC in 15.1%. Half (49.1%) of the upregulated unigenes had homologous hits in the Swiss-Prot database.

At 1 HPI, several unigenes with documented or putative immune functions were differentially expressed with VSNJV infection (Table 1, Figure 4). Specifically, *SMAD6*, *TOLL7*, *SAM11*, and *KEN1* were all downregulated −2.3 and −0.8 Log2FC or −4.8 to −1.8-fold change with VSNJV infection. Proviral unigene *KI26L* was also downregulated an average of −0.84 Log2FC between two unigenes. Several non-immune unigenes were downregulated with VSNJV infection, such as the circadian rhythm regulator *REG5* (−2.4 LogFC) [37,38] and age regulator *LIPT* (average −1.35 Log2FC) [39]. Multiple immune unigenes were upregulated in response to VSNJV infection at 1 HPI, including *TRAF4*, *PPAF3*, *CHIT1*, *FAT*, and *ABCA3* with Log2FC values ranging from 0.8 to 2.4 (Table 1). *ATG2B*, which has possible immune functions and proviral functions, was upregulated 0.83 Log2FC.

### 3.3. VSNJV Transcriptome from W8 Cells

Transcription of VSV proteins occurs sequentially starting with the N gene [57,58]. Furthermore, transcription is discontinuous and pauses at the regions between genes (Figure 5A), possibly as the RNA polymerase complex stutters or falls off [59]. At 1 HPI, the number of unigenes by viral gene followed the expected linear decrease across the genome with N being the most numerous unigene and L the least numerous (Figure 5B, Appendix A). Indeed, gene order was negatively correlated with unigenes per million at 1 HPI (r (28) = −0.93, *p* = 0.02) but not at any time point after. Across all time points, the number of P unigenes per million remained steady, while after 1 HPI N decreased and G, M, and L all increased (Figure 5B). Overall, the least frequent viral unigene was L, the last and largest gene. The biggest difference in unigene amount, for all genes except P, occurred between 1 and 8 HPI.

## 4. Discussion

The transcriptomic profiles of *C. sonorensis* W8 cells in response to VSNJV infection at multiple time points indicated that unigenes in the W8 cells only were differentially expressed at 1 HPI but not at 8, 24, or 96 HPI. The activity of the VSNJV matrix protein could contribute to the time-dependent differences in *C. sonorensis* transcriptomic response. The matrix protein is integral for VSV replication, virion packaging, virus budding, and evasion of the host antiviral response [60]. The latter is accomplished when the matrix suppresses the transcription of host innate immune genes and blocks host mRNA transport out of the nucleus [60,61,62,63,64,65,66,67,68]. In the current study, the number of matrix unigenes is lowest at 1 HPI compared with the later time points, suggesting that transcription of the matrix specifically may suppress host transcriptomic response. In such a scenario, the lack of response to the virus during robust replication may reveal our first insights into mechanisms of viral tolerance, and therefore permissiveness and competence in *C. sonorensis* that warrant further exploration via in vivo studies. Additionally, the lack of differential expression at the later time points in *Culicoides* cells could be related to VSV’s lysogenic replication strategy in *Culicoides* cells that does not cause apoptosis [9].

Several of the differentially expressed W8 unigenes have known immune functions or are thought to be components of immune system pathways. The upregulation and downregulation of immune genes during viral infection have been documented with other arthropod-borne viruses, such as dengue virus [15], Zika virus [69], and epizootic hemorrhagic disease virus (EHDV) [20], illustrating how complex the relationship is between vector and virus. In the current study, *TOLL-7*, a receptor involved with VSV binding and transduction of antiviral immune signaling and autophagy in *Drosophila* [41], was downregulated almost two-fold with VSNJV infection. While VSNJV infection appears to possibly suppress autophagy via the downregulation of *TOLL-7*, the virus also upregulated *ATG2B*, a protein required for autophagy [70], at almost the same levels that *TOLL-7* was suppressed. Simultaneous up- and downregulation of autophagy in arbovirus-infected insect cells has been previously reported [48]. Interestingly, *SMAD6* and *KEN1*, negative regulators of the transforming growth factors (TGF)-β [71] and the JAK/STAT pathway [46], respectively, were both suppressed following VSNJV infection in W8 cells. By suppressing these genes during infection, W8 cells appear to be increasing their inflammatory pathways 1 HPI, but not after. *TRAF4* inhibits activation of both Toll-like receptor-mediated NK-κβ and interferon, making *TRAF4* the only member of the *TRAF* family to negatively regulate immune signaling [72,73,74]. The current data suggest that in W8 cells, VSNJV infection upregulates *TRAF4* 1.7-fold, which would further suppress the immune response in the infected cells. Kinesin-1 is a motor protein that viruses utilize for intracellular movement and disruption of the viral capsid during entry [43,44,75]. Das et al. [76] postulated that the VSV nucleocapsid interacts with kinesin-1 to move along microtubule tracks within the cell. The data imply that W8 cells suppress kinesein-1-mediated viral movement as the expression of *KI26L* decreased 1.8-fold compared with the control cells.

*ABCA3* is a member of the ABC transporter gene family, which could be involved in the mosquito immune response to viral infection [55]. However, while many of the ABC genes were found to be differentially expressed in mosquitoes following dengue, yellow fever, and West Nile virus infection, *ABCA3* specifically was not identified so its potential antiviral role is unknown. Expression of *ABCA3* in VSNJV-infected W8 cells ranged from 2.7- to 5.4-fold higher than control cells, suggesting that unigene plays a role in VSNJV infection. Future studies on the ABC transporter gene in the context of VSV infection are warranted. In *Drosophila*, *FAT* is a receptor of the Hippo tumor-suppressor pathway that controls cellular growth, migration, and survival [53]. In the current study, VSNJV infection increased the expression of *FAT* 3.4-fold higher than the uninfected cells. The Hippo pathways are also known to have antibacterial functions [53], but to the best of our knowledge have not been studied in the context of viral infection. Two other unigenes, *PPAF3* and *GGT*, were upregulated with VSNJV infection in W8 cells, and both are thought to play a role in the activation of the innate immune response [50,56]. Additionally, we identified several unigenes that have no known immune or antiviral functions that require further study for their putative role in midge–virus interactions.

The midge transcriptomic response to arboviral infection has been tested previously in vivo in female *C. sonorensis* who were fed EHDV and examined 36 h post ingestion [20]. There are clear caveats to be considered when comparing the results of that study to the one presented here due to the differences in the viruses and the hosts within which responses were analyzed (here, in vitro embryonic cells vs. in vivo whole midges). Nonetheless, such comparisons could provide insight into the molecular underpinnings of vector–virus interactions in the midge. Of the 49 unigenes that were downregulated 1 h after infection with VSV in this study, 36 (~74%) were also downregulated in whole midges at 36 h post ingestion/infection with EHDV. One unigene, *REG5*, an ortholog to *dreg-5* (*Drosophila* rhythmically expressed gene 5) in *D. melanogaster* [38] was downregulated about four-fold after virus infection in both studies. This gene is controlled by, and in phase with, the circadian timekeeper gene *period*, and is putatively involved with a yet-defined circadian process in insects [37,38]. Interestingly, the major response to EHDV infection in whole midges at 36 h was downregulation of >1400 unigenes (~60% of total differentially expressed unigenes) and 122 were associated with neuro-sensory behaviors, including not only *REG5* but also the circadian rhythm regulators *clock*, *cycle*, and numerous target genes [20]. Another intriguing finding from the current study was a lack of any detectable upregulation of antiviral defense pathways or effectors in midge cells after VSV infection. In EHDV-infected female midges, unigenes coding for antiviral defense signaling, such as two *dome* and four *toll* paralogs, were all downregulated [20]. Interestingly, in both studies, unigenes for *TOLL-7* were downregulated with virus infection. In the current study, of the 53 unigenes upregulated during VSV infection at 1 h, 24 had no corresponding unigene in the EHDV reference transcriptome. Of the remaining 29 upregulated unigenes, 5 were significantly upregulated and 5 significantly downregulated in the EHDV study. We can infer that these results indicate that the positive transcriptional response to the two viruses (VSV vs. EHDV) in the two systems tested (cell line vs. whole midge) is where biological variability exists. Reciprocal studies of the transcriptome response of midge cell lines to EHDV and of whole midges to VSV certainly warrant further investigation in the future.

This study provides the first insights into the molecular–genetic interactions between VSV and cells of one of its arthropod vectors, *C. sonorensis*. On a transcriptional level, midge cells only showed a response to virus infection at 1 h post infection. Extrapolating these findings to understand what may occur between midges and VSV in vivo is challenging. However, this study is a first step toward understanding the *Culicoides* transcriptomic response to VSV infection, or potential lack thereof, that may underlie their permissiveness as vectors for this important zoonotic virus. A further understanding of the molecular interactions between VSV and midges, including protein-level expression, should be incorporated into future studies to fully elucidate the complex relationship between VSV and *Culicoides* biting midges.

## Figures and Tables

**Figure 1 viruses-15-02108-f001:**
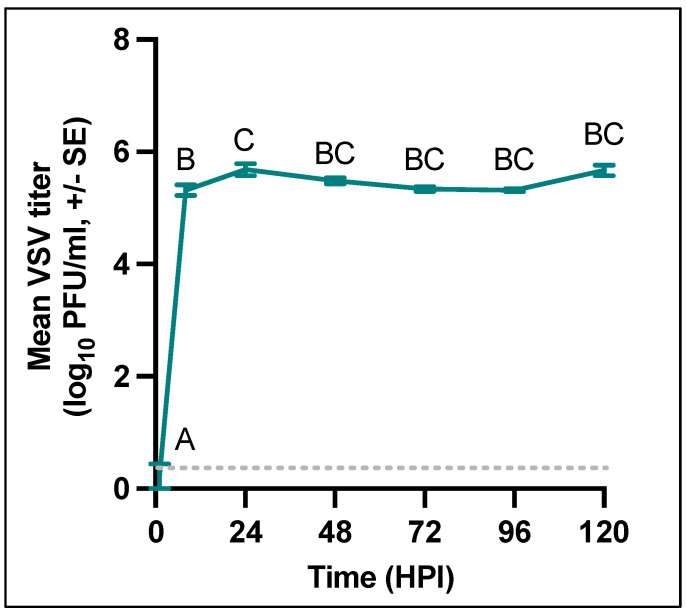
(A) One-step growth curve of VSNJV in W8 cells (n = 6 replicates per time point, repeated measures ANOVA F (6,41) = 361.7, *p* < 0.0001). The dotted line indicates the limit of detection. Groups that do not share a letter are significantly different via Tukey pairwise comparison (*p* < 0.05).

**Figure 2 viruses-15-02108-f002:**
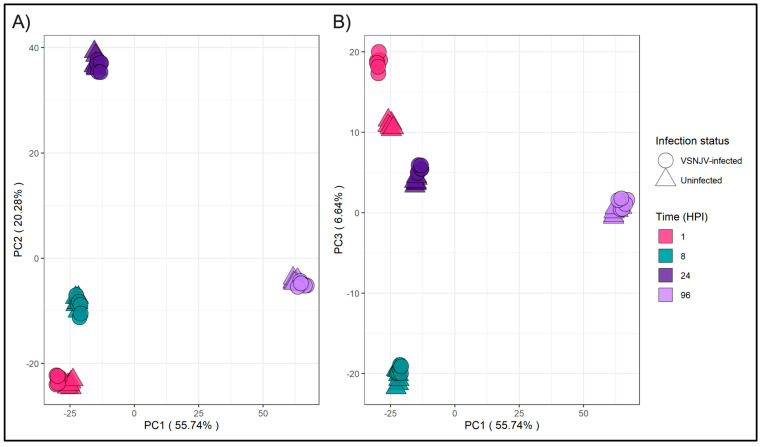
Principal component analysis (PCA) of transcriptomic data. (**A**) Score plot of PC1 vs. PC2. (**B**) Score plot of PC1 vs. PC3. Shapes indicate infection status and color indicates the time point.

**Figure 3 viruses-15-02108-f003:**
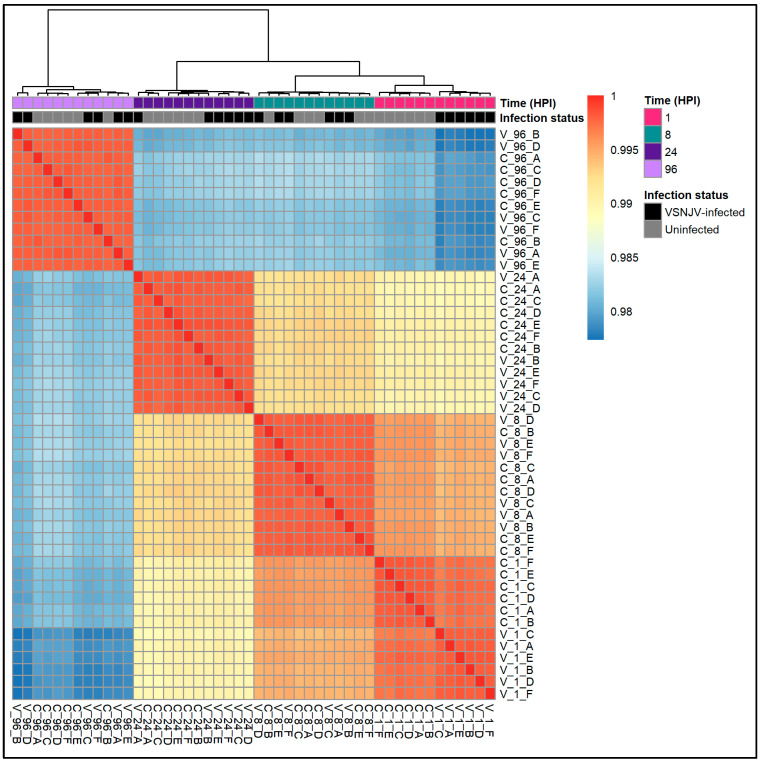
Heatmap of sample correlation based on regularized log (rlog)-transformed gene expression data clustered by correlation distance. Color intensity indicates the strength of pairwise Pearson correlation coefficients between all samples, with red indicating a higher positive correlation and blue indicating a lower correlation. Samples are labeled with V for VSNJV-infected cells or C for mock-infected controls followed by the collection time point (HPI) and letter (A–F) to indicate the replicate (n = 6 replicates per time point and condition).

**Figure 4 viruses-15-02108-f004:**
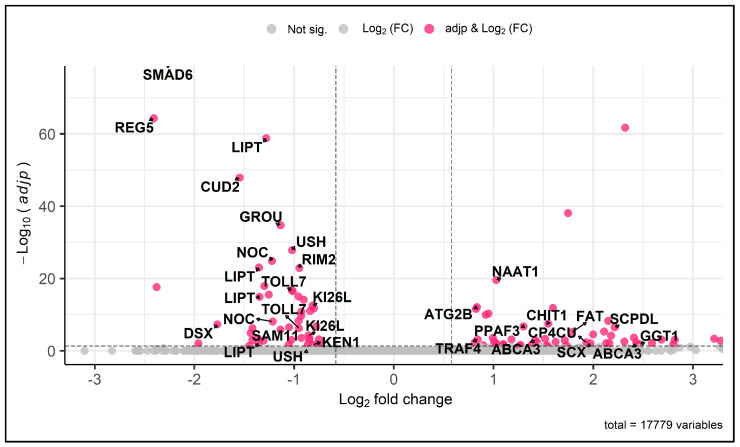
Enhanced volcano plot of differential unigene expression between VSNJV-infected and mock-infected *C. sonorensis* W8 cells 1 HPI. The volcano plot shows the relationship between fold change (*x*-axis) and statistical significance (*y*-axis) in determining differentially expressed genes between virus-treated and control cells at time point 1 HPI. Each dot represents a unigene, with pink dots indicating unigenes meeting the significance threshold of adjusted *p*-value < 0.05 and biological relevance cutoff of log2 (fold change) > 0.58 (1.5-fold change). Differential expression analysis was performed using DESeq2 on RNA sequencing data, with statistical testing adjusted for a single factor using the Wald hypothesis test.

**Figure 5 viruses-15-02108-f005:**
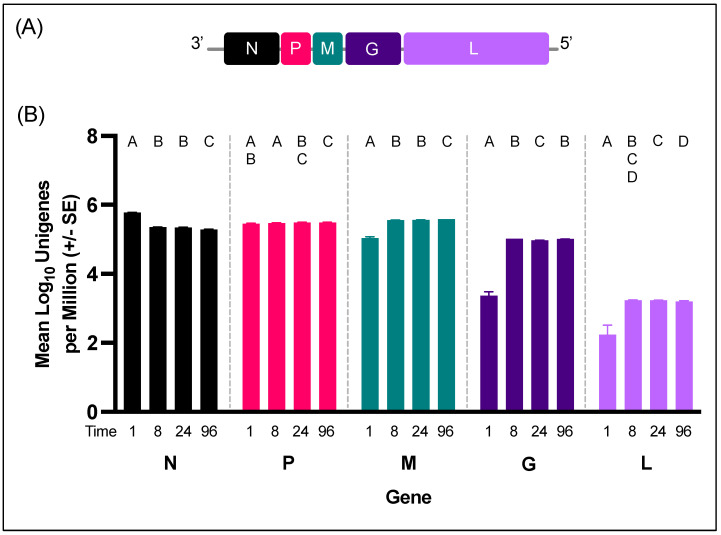
VSNJV unigenes from W8 infection. (**A**) Diagram of VSNJV genome. (**B**) Mean Log_10_ VSNJV unigenes per million at each sequenced time point (n = 6 replicates per time point, two-way ANOVA F (12,75) = 903.6, *p* < 0.0001). Colors indicate viral genes (black: N, pink: P, teal: M, dark purple: G, and light purple: L). Within each gene, time points that do not share a letter are significantly different via Tukey pairwise comparisons (*p* < 0.05). Full pairwise comparisons can be found in Appendix A.

**Table 1 viruses-15-02108-t001:** Select differentially expressed unigenes with Log2 FC > 0.8 or −0.8.

Functional Annotation	Name	Log2 FC	FDR *p*-Value	Possible Immune or Proviral Function
REG5	Rhythmically expressed gene 5	−2.41	4.72 × 10^−65^	Unknown
SMAD6	Mothers against decapentaplegic homolog 6	−2.27	5.23 × 10^−266^	Immune [40]
DSX	Protein doublesex	−1.77	4.88 × 10^−8^	Unknown
CUD2	Endocuticle structural glycoprotein	−1.55	1.23 × 10^−48^	Unknown
LIPT	Lipoyltransferae 1 *	−1.35, −1.36, −1.35	1.15 × 10^−15^, 0.007, 8.35 × 10^−24^	Unknown
TOLL7	Toll-like receptor 7 *	−1.02, −0.95	1.64 × 10^−17^, 5.17 × 10^−7^	Immune [41]
SAM11	Sterile alpha motif domain-containing protein 11	−0.93	2.54 × 10^−4^	Immune [42]
KI26L	Kinesin-like protein *	−0.82, −0.85	3.00 × 10^−13^, 5.10 × 10^−5^	Proviral [43,44]
KEN1	Transcription factor Ken1	−0.81	0.02	Immune [45,46]
ATG2B	Autophagy-related protein 2 homolog B	0.83	7.75 × 10^−13^	Immune, proviral [47,48]
TRAF4	TNF receptor-associated factor 4	0.84	6.59 × 10^−4^	Immune [49]
NAAT1	Sodium-dependent nutrient amino acid transporter 1	1.03	2.53 × 10^−20^	Unknown
PPAF3	Phenoloxidase-activating factor 3	1.30	1.73 × 10^−7^	Immune [50]
CHIT1	Chitotriosidase 1	1.55	3.03 × 10^−8^	Immune [51,52]
FAT	Cadherin-related tumor suppressor/FAT tumor suppressor homolog 1	1.78	6.51 × 10^−6^	Immune [53,54]
CP4CU	Cytochrome P450 4c21	1.93	0.005	Unknown
SCX	Basic helix-loop-helix transcription factor scleraxis	1.97	0.009	Unknown
ABCA3	Phospholipid-transporting ATPase ABCA3	2.44, 1.43	0.02, 3.72 × 10^−6^	Immune [55]
GGT	Gamma-glutamytranspeptidase 1	2.47	0.02	Immune [56]

* Multiple unigenes.

## Data Availability

Transcriptome sequencing data are available at NCBI study accession SRP461227, within Bioproject accession: PRJNA101813. Analysis scripts are available at doi.org/10.5281/zenodo.8342149.

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
