# Peer review of "Vesicular Stomatitis Virus Elicits Early Transcriptome Response in Culicoides sonorensis Cells"

_viruses, 2023, doi:10.3390/v15102108_

Round 1

Reviewer 1 Report

Overall this is a well written paper describing some very basic observations of gene expression in cell culture of an arthropod.  I believe it will be useful for the next steps of testing these observations in intact insects or in observing different results in incompetent vectors.

Lines 31-32. This is not factually true. VS is similar to FMD but they can be distinguished. For example, any horse cases are immediately distinguishable. I would phrase this part to say they can be mistaken for each other and for other vesicular diseases.

Line 34: This is subject to local, state, territory or national regulations. I would say “can be” placed under quarantine.

Line 36: should that be “Culicoides sonorensis” s.l. or s.s.?

Line 45: “ like mosquitoes” perhaps should be “such as mosquitoes”

Line 49: I think you are safer saying “whether these effectors were aimed at the virus”

Line 51 or 62: It might be very helpful to describe the cell type for C. sonorensis W8 cells.  Are these gut, ovarian, muscle, etc cells or are they of unknown origins.

Line 141-149 read like a summary of the methods.  Feel free to keep them here or move them to methods and just present the data on the results.

Line 168: Is “deomstrate” a word?

Line 199-209: This might be semantic but do you know this is post infection or in the process of infection. I say this because you are stating that something is occurring at 1 hour post infection however these gene expression changes could be in response to the viral stress as cells try to repel infection and thus are not infected yet just post inoculation.

Line 241-256: Same statement as above.

Line 327: Delete “Interestingly”

Author Response

Overall this is a well written paper describing some very basic observations of gene expression in cell culture of an arthropod.  I believe it will be useful for the next steps of testing these observations in intact insects or in observing different results in incompetent vectors.

Lines 31-32. This is not factually true. VS is similar to FMD but they can be distinguished. For example, any horse cases are immediately distinguishable. I would phrase this part to say they can be mistaken for each other and for other vesicular diseases.

We have corrected the wording of this sentence.

Line 34: This is subject to local, state, territory or national regulations. I would say “can be” placed under quarantine.

Done

Line 36: should that be “Culicoides sonorensis” s.l. or s.s.?

That nomenclature does not apply to this situation. We checked the literature, and we are up to date on the taxonomic nomenclature.

Line 45: “ like mosquitoes” perhaps should be “such as mosquitoes”

Done

Line 49: I think you are safer saying “whether these effectors were aimed at the virus”

Done

Line 51 or 62: It might be very helpful to describe the cell type for C. sonorensis W8 cells.  Are these gut, ovarian, muscle, etc cells or are they of unknown origins.

We clarified the origin of the cells in section 2.1. They were generated using embryonated eggs.

Line 141-149 read like a summary of the methods.  Feel free to keep them here or move them to methods and just present the data on the results.

Thank you for your comment and we agree that there was too much repeated detail. We removed some of the summary details and moved the line justifying the time points selected to the methods section 2. We left one line describing the experiment to orient the reader at the beginning of the results section.

Line 168: Is “deomstrate” a word?

We have fixed the spelling error

Line 199-209: This might be semantic but do you know this is post infection or in the process of infection. I say this because you are stating that something is occurring at 1 hour post infection however these gene expression changes could be in response to the viral stress as cells try to repel infection and thus are not infected yet just post inoculation. Line 241-256: Same statement as above.

We were using “Post-infection” to refer to the time after we added the viral inoculum to the cells. But the reviewer is correct that the time the viral inoculum is added is not necessarily the actual infection time. We changed hours post infection to hours post inoculation to clarify the mechanical nature of the definition and not the biological implication of infection.

Line 327: Delete “Interestingly”

Done

Reviewer 2 Report

In their manuscript, Scroggs and colleagues have characterized the transcriptomic responses of VSV-infected and mock infected Culicoides sonorensis cells at 1, 8, 25 and 96h post infection.

They observe a significant transcriptomic response in VSV-infected cells at 1h p.i. while at 8, 24, and 96h p.i. there is no difference between the transcriptomic profiles of VSV-infected and mock infected cells. This is quite unexpected and not easy to explain.  

Globally, the work is primarily descriptive. The authors observe that some genes involved in immunity are either upregulated (ATG2B and 17 TRAF4) or downregulated (SMAD6 and TOLL7) in VSV-infected cells at 1h p.i. The authors conclude that VSV infection in midge cells produces an early immune response that quickly wanes. It is extremely difficult to go further in interpreting the data and even to interpret the real signification of this response.

These are therefore preliminary results. However, we can consider that these data are of interest to the community working on the interaction between VSV and its insect vectors.    

Major points

1) In figure 1 the authors claim that their data correponds to multicycle replication kinetics of VSNJV in W8 cells. This is not correct. As the MOI=5 (as indicated in the material and methods), these are one-step growth curves. All the cells are infected at the same time and there is no second cycle.

2) After 8h, the maximum titer is reached (the difference between 8 and 24h p.i. is very weak and probably not biologically significant; indeed, there is no significant difference for the data obtained at later times neither with those obtained at 8h p.i.. nor with those obtained at 24 h p.i.). This means that after 8h, no or only a few virions are released in the cell culture medium. The claim made in the abstract that viral replication remains high after 8h p.i. is therefore doubtful.

3) From reading the manuscript, it is not clear whether the transcriptomic data were obtained from multiple independent biological replicates.

4) There are apparently several differences between VSV infection in mammalian cells and in Culicoides sonorensis. In mammalian cells, as mentioned by the authors, VSV M protein interferes with mRNA export from the nucleus (this does not appear to be the case in Culicoides sonorensis). In mammalian cells, VSV infection induces a strong cytopathic effect accompanied by cell rounding. Reference 9 indicates that it is not the case throughout the whole organism of Culicoides sonorensis. It would be good to check if this is indeed the case in W8 cells (by immunofluorescence using a polyclonal antibody against VSV, or a specific monoclonal antobody against a viral protein).

5) Also, it would be good to have an idea of the time line of the replication cycle (time of primary/secondary transcription, replication, and viral budding). I would suggest performing RTqPCR to quantify VSV mRNAs, genomes and antigenomes at 2h, 4h and 6h p.i. Finally, viral production could also be characterized at 2, 4 and 6h p.i.    

The experiments suggested in points 4 and 5 could reinforce the manuscript.     

English is good although a few typos remain in the text. 

Author Response

In their manuscript, Scroggs and colleagues have characterized the transcriptomic responses of VSV-infected and mock infected Culicoides sonorensis cells at 1, 8, 25 and 96h post infection.They observe a significant transcriptomic response in VSV-infected cells at 1h p.i. while at 8, 24, and 96h p.i. there is no difference between the transcriptomic profiles of VSV-infected and mock infected cells. This is quite unexpected and not easy to explain.  Globally, the work is primarily descriptive. The authors observe that some genes involved in immunity are either upregulated (ATG2B and 17 TRAF4) or downregulated (SMAD6 and TOLL7) in VSV-infected cells at 1h p.i. The authors conclude that VSV infection in midge cells produces an early immune response that quickly wanes. It is extremely difficult to go further in interpreting the data and even to interpret the real signification of this response.These are therefore preliminary results. However, we can consider that these data are of interest to the community working on the interaction between VSV and its insect vectors.    

Major points

1) In figure 1 the authors claim that their data corresponds to multicycle replication kinetics of VSNJV in W8 cells. This is not correct. As the MOI=5 (as indicated in the material and methods), these are one-step growth curves. All the cells are infected at the same time and there is no second cycle.

We have corrected the figure legend.

2) After 8h, the maximum titer is reached (the difference between 8 and 24h p.i. is very weak and probably not biologically significant; indeed, there is no significant difference for the data obtained at later times neither with those obtained at 8h p.i.. nor with those obtained at 24 h p.i.). This means that after 8h, no or only a few virions are released in the cell culture medium. The claim made in the abstract that viral replication remains high after 8h p.i. is therefore doubtful.

We have removed the line from the abstract and edited the results section 3.1 for clarity.

3) From reading the manuscript, it is not clear whether the transcriptomic data were obtained from multiple independent biological replicates.

We are sorry that this wasn’t clear in the manuscript. We included 6 biological replicates for the VSV infected cells and 6 biological replicates for the mock infected cells to account for potential variation in the transcriptome data. Lines 87-88 and 157 include this information. We also added the information to the legends of Figure 1, Figure 3, and Figure 5.  

4) There are apparently several differences between VSV infection in mammalian cells and in Culicoides sonorensis. In mammalian cells, as mentioned by the authors, VSV M protein interferes with mRNA export from the nucleus (this does not appear to be the case in Culicoides sonorensis). In mammalian cells, VSV infection induces a strong cytopathic effect accompanied by cell rounding. Reference 9 indicates that it is not the case throughout the whole organism of Culicoides sonorensis. It would be good to check if this is indeed the case in W8 cells (by immunofluorescence using a polyclonal antibody against VSV, or a specific monoclonal antobody against a viral protein).

5) Also, it would be good to have an idea of the time line of the replication cycle (time of primary/secondary transcription, replication, and viral budding). I would suggest performing RTqPCR to quantify VSV mRNAs, genomes and antigenomes at 2h, 4h and 6h p.i. Finally, viral production could also be characterized at 2, 4 and 6h p.i.    

The experiments suggested in points 4 and 5 could reinforce the manuscript.   

We thank the reviewer for their feedback in points 4 and 5. We agree that the stages of the VSV life cycle in W8 cells should be studied in detail and compared to mammalian cells and that the experiments from reference 9 can be expanded to include W8 cells to further elucidate the impacts of VSV infection in W8 cells compared to mammalian cells. early in infection and the differences between infection in C. sonorensis and mammalian cells. We plan to evaluate these further, but given the revision deadline of 10 days, these experiments will be included in a future manuscript.

Reviewer 3 Report

Scroggs et al., report the findings of a transcriptome analysis of Vesicular stomatitis virus infection of Culicoides sonorensis W8 cells. Results show an early immune response that is absent in the later stages of virus infection promoting permissivenes of the cell line. Furthermore, the genes that were differentially expressed are identified. 

The study is concise and results are presented clearly, and clusions are drawn according to the results presented. 

there are few spelling mistakes that needs to be corrected especially in section 3.2 lines 155 to 174.

Author Response

Scroggs et al., report the findings of a transcriptome analysis of Vesicular stomatitis virus infection of Culicoides sonorensis W8 cells. Results show an early immune response that is absent in the later stages of virus infection promoting permissivenes of the cell line. Furthermore, the genes that were differentially expressed are identified. 

The study is concise and results are presented clearly, and conclusions are drawn according to the results presented. 

There are few spelling mistakes that needs to be corrected especially in section 3.2 lines 155 to 174.

Thank you for your review. We have corrected the typos and spelling mistakes in the manuscript.